

# Sub-millennial climate variability from high resolution water isotopes in the EDC ice core

Antoine Grisart[1], Mathieu Casado[1,3], Vasileios Gkinis[2], Bo Vinther[2], Philippe Naveau[1], Mathieu Vrac[1], Thomas Laepple[3], Bénédicte Minster[1], Frederic Prié[1], Barbara Stenni[4], Elise Fourré[1], Hans-Christian Steen Larsen[5], Jean Jouzel[1], Martin Werner[6], Katy Pol[1], Valérie Masson-Delmotte[1], Maria Hoerhold[6], Trevor Popp[2], Amaelle Landais[1]

[1] Laboratoire des Sciences du Climat et de l'Environnement, CEA–CNRS–UVSQ–Paris-Saclay–IPSL, Gif-sur-Yvette, France
[2] Physics of Ice, Climate and Earth, Niels Bohr Institute, University of Copenhagen, Copenhagen, Denmark
[3] Alfred Wegener Institute, Helmholtz Center for Polar and Marine Research, Potsdam, Germany
[4] Department of Environmental Sciences, Informatics and Statistics, University Ca' Foscari of Venice, Venice, Italy
[5] Geophysical Institute, University of Bergen and Bjerknes Centre for Climate Research, Bergen 5020, Norway
[6] Alfred Wegener Institute, Helmholtz Centre for Polar and Marine Research, Bremerhaven, Germany

*Correspondence to*: Antoine Grisart (antoine.grisart@lsce.ipsl.fr)

**Abstract.** The EPICA Dome C (EDC) ice core provides the longest continuous climatic record covering the last 800 000 years (800 kyrs). Obtaining homogeneous high resolution measurements and accounting for diffusion provide a unique opportunity to study decadal to millennial variability within the past glacial and interglacial periods. We present here a compilation of high resolution (11 cm) water isotopic records with 27,000 $\delta^{18}O$ measurements and 7,920 $\delta D$ measurements (covering respectively 94 % and 27 % of the whole EDC record), including both published and new measurements (2,900 for both $\delta^{18}O$ and $\delta D$) over the last 800 kyrs on the EDC ice core. We show that overlapping time series performed over multiple depth ranges over the past 20 years, using different analytical methods and in different laboratories, are consistent within analytical uncertainty, and therefore can be combined to provide a homogeneous data set. A frequency decomposition of the most complete $\delta^{18}O$ record and a simple assessment of the possible influence of diffusion on the measured profile shows that the variability during glacial periods at multi-decadal to multi-centennial timescale is higher than variability of the interglacial periods. This analysis shows as well that during interglacial periods characterized by a temperature optimum at its beginning, the multi-centennial variability is the strongest over this temperature optimum.

## 1 Introduction

Water isotopes in ice cores ($\delta^{18}O$, $\delta D$) are valuable tools to reconstruct past temperatures in polar regions. Along air mass transportation, distillation of moisture from the low latitude regions of evaporation to the polar regions leads to a loss of heavy isotopes during successive precipitation events and hence to a decrease of $\delta^{18}O$ and $\delta D$ toward cold regions. Despite known limitations due to temporal changes in intermittency of precipitation (Casado et al., 2020), vapor origin and transport (Helsen et al., 2006), sea ice extent (Noone, 2004), changes in condensation vs surface temperatures (Buizert et al., 2021) or deposition and post-deposition effects (Casado et al., 2018), the spatial relationship between surface temperature and surface snow $\delta D$





and $\delta^{18}O$ has long been used to establish an isotopic paleothermometer to infer past temperature variations at least qualitatively (Jouzel et al., 2013).

Today, the oldest continuous isotopic record from ice cores has been retrieved at the Dome C through the European Project for Ice Coring Antarctica (EPICA) Dome C ice core (EDC) covering the last 800,000 years (800 kyrs) (Jouzel et al., 2007). The first analyses of water isotopic composition (δD) over the EDC ice core were displayed at a ~4 m resolution providing the first picture of the succession of the 8 glacial – interglacial cycles (EPICA community members, 2004). Several years later, systematic measurements of δD on bag samples (55 cm) evidenced the millennial scale variability over the glacial periods in

Antarctica (Jouzel et al., 2007; Stenni et al., 2010). In the following years, some studies focused on even higher resolution (11 cm) on some key periods to study the high frequency climate variability. In order to explore potential changes in high frequency variability in between different interglacial periods, Pol et al., (2010, 2011, 2014) used 11 cm resolution δD measurements over interglacial periods during Marine Isotopic Stages (MIS) 5, 11 and 19, i.e. the periods between 112 and 134 ka (before present), 392 and 427 ka, and 747 and 800 ka, respectively. Landais et al., (2015) focused on 11 cm resolution $\delta^{18}O$ over the

last glacial period back to 60 kyrs.

It is challenging to retrieve the absolute decadal variability from central Antarctic records (Ekaykin et al., 2017; Casado et al., 2020). But since the processes affecting the signal should not vary too much in interglacial conditions, by comparing interglacial periods MIS5 and 11 enabled to estimate the relative variations of decadal to centennial climate variability with respect to the Holocene's (Pol et al., 2011; 2014). Over the last glacial period, the high resolution $\delta^{18}O$ record showed an

enhanced amplitude of the multi-decadal to centennial variability during the warm phases of the Antarctic Isotopic Maxima, or AIM (Landais et al., 2015). These AIM events are key climatic features of the last glacial period: they are counterparts of the Northern Hemisphere abrupt temperature increases first identified in the Greenland ice cores (Dansgaard et al., 1985; Blunier and Brook, 2001; EPICA community members, 2006).

High resolution water isotopic measurements over the EDC ice core are hence key to document the temporal patterns of

climatic variability over the past 800 kyrs. Unfortunately, the analytical load to obtain the full 800 kyrs record at 11 cm resolution is enormous, and would represent 35,000 measurements. Even if several individual studies have been published, a complete synthesis of EDC high resolution δD and $\delta^{18}O$ records over the last 800 kyrs is still missing. This is an important limitation for the documentation of past changes in sub-orbital climatic variability in Antarctica and to compare the climatic variability features between glacial and interglacial periods or between different interglacial (glacial) periods. As an example,

while we know that the interglacial periods before the Mid-Brunhes Transition (MBT, 430 ka) are cooler than the five most recent interglacial periods, we lack documentation of the high resolution climate variability during interglacial periods before and after the MBT (Barth et al., 2018; Past Interglacials Working Group of PAGES, 2016). A first challenge is thus to provide homogeneous high resolution isotopic records.

A second challenge to characterize the past high frequency climate variability in Antarctica is non-temperature related

variability in the water isotopes from the depositional process (;  Laepple, 2018) and smoothing / filtering effects of post-deposition processes. Indeed, post-deposition processes (Casado et al., 2020, 2018) and firn and ice diffusion (Gkinis et al.,



2011, 2021) strongly limit the interpretation of water isotopic variability in term of climatic variability. In the case of old ice, the impact of diffusion which increases with depth and age can reach multi-centennial time scales and affect the climate variability recorded in $\delta^{18}O$ and $\delta D$. Pol et al. (2010) showed that the 11 cm resolution $\delta D$ record of MIS 19 (3147 – 3190 m deep in the EDC ice core) was not bringing more information than the 55 cm resolution record because of large impact of diffusion at this depth of the core. This effect is particularly important to quantify for the 1.5 Ma ice core to be drilled in East Antarctica. Indeed, documenting the evolution of diffusion length with depth is key to anticipate what kind of information on climate variability can be retrieved from the deepest part of this future ice core.

Here we address the two aforementioned challenges (high-resolution records and influence of diffusion) by presenting a compilation of new high resolution measurements of $\delta^{18}O$ and $\delta D$ on the EDC ice core. The first section presents the analytical methods used to perform high resolution measurements of $\delta D$ and $\delta^{18}O$ of the different sections of the EDC ice core over the last decades as well as methods for spectral analyses and calculation of isotopic diffusion along the EDC ice core. The second section describes how the different measurements performed over the past 20 years in different institutes with different analytical methods can be compiled together into a single record. The third section uses the high resolution measurements to investigate changes in sub-orbital climatic variability across the last 800 kyrs and how diffusion can affect some of the observed features.

## 2. Materials and methods

### 2.1 The EPICA ice core

The Concordia Franco-Italian station is located at 3 233 m above sea level on the continental plateau of Antarctica, (75°06′12″S 123°21′30″E). The mean annual surface temperature is -54.5°C and the snow accumulation rate is around 25 mm water equivalent.yr$^{-1}$ (EPICA community members, 2004; Le Meur et al., 2018).

The EDC ice core has been drilled at the Concordia station on the Dome C where the ice was supposed to be the less deformed. The drilling project was conducted over the period 1996 – 2004. In 1999, the drill was stuck at 788 m depth (45 ka) so a new drilling began from the surface the same year a few meters away from the first drilling hole. The bedrock was then successfully reached in 2004 at a depth of 3190 m. This second ice core is referred as EDC2. By the time this second ice core was retrieved, the full 788 meters of EDC1 were analysed. Later, EDC2 measurements started 19 meters higher than the bottom end of the EDC1 ice core in order to have an overlap to reconnect the two cores without duplicating all the measurements on the common depth range.

After drilling and logging, the ice core was cut in 55 cm long parts. 55 cm sections were then cut longitudinally on site for several measurements (water isotopes, physical properties, $^{10}Be$, chemistry, gas). An archive piece (~ one quarter of the section) is stored in polystyrene boxes in the EPICA snow-cave at the Concordia station at -50°C. Two types of samples were dedicated for the continuous analyses of water isotopes on the EDC ice core. First, a 55 cm long stick with a 1 cm$^2$ cross section was melted and stored on site in plastic bottles for the low resolution measurements. Another section (stick with 2*1 cm cross





section) was cut in 5 parts of 11 cm each for the high resolution measurements and placed in plastic bags, stored at -20°C
during a few months before being melted and transferred into plastic bottles kept at -20°C.

## 2.2 Measurements techniques and coherency of the dataset

Over the last 20 years, several techniques have been applied to measure $\delta D$ and $\delta^{18}O$ of the EDC ice core (Tables 1 and 2).
The first measurements on the first EDC ice core (DC-96) were performed using the uranium reduction method for $\delta D$ (Vaughn
et al., 1998) or the $CO_2 – H_2O$ equilibration method for $\delta^{18}O$ (Meyer et al., 2000). The latest measurements were performed
on the EDC2 ice core using a method based on cavity ring down spectroscopy (CRDS) (Kerstel and Gianfrani, 2008; Busch
and Busch, 1999). The precisions obtained for the different methods are comparable, i.e. 2 $\sigma$ values between 1 and 1.4 ‰ for
$\delta D$ and between 0.1 and 0.4 ‰ for $\delta^{18}O$ (Table 2). Figure 1 displays the full high resolution (11 cm) datasets for $\delta^{18}O$ and $\delta D$
of water over the EDC ice core (figure 1).

## 2.3 The EPICA ice core

A discrete wavelet analysis is used to identify the contribution to the overall isotopic variability from signals of different
periodicities (i.e. corresponding to decadal to multi-millennial signal variability). With this aim, we produced a multi resolution
analysis (MRA) using R software with the waveslim wavelet package (Whitcher, 2020) containing the MRA function with a
Daubechies orthonormal wavelet filter. MRA is a mathematical analysis tool which decomposes a signal at different resolution
levels. An important feature of MRA is its ability to capture temporally localised changes at its nearest neighbour.  A low
(high) resolution level corresponds to a coarse (detailed/high frequency) component of the original signal. Each MRA level
can thus be used to interpret the temporal variability within a frequency range. Adding all MRA levels exactly reproduce the
original undecomposed signal. The wavelet analysis needs to be applied on time intervals with a uniform resolution. Because
we aim to keep as much as possible information on the climatic variability inferred from the high resolution isotopic
measurements, the EDC record expressed on the AICC2012 age scale has been split in 6 intervals with decreasing resolution
from the top (youngest section between 0 and 56 ka where 11 cm corresponds to a 10 yr resolution on the AICC2012 age
scale) to the bottom of the core (oldest section between 651 and 800 ka where 11 cm correspond to a 320 yr resolution on the
AICC2012 age scale). The decomposition is explained on Table 3.

The resolution of the MRA for the different intervals was chosen to increase by a factor of two between two neighbouring
intervals i and i+1, i being a number between 1 and 5. As a consequence, the 2$^{nd}$ MRA decomposition of the interval i has the
same resolution than the 1$^{st}$ MRA of the interval i+1 (Table 3). We then concatenate the MRA with the same temporal
resolution, leading to 9 successive composites (named a, b, c, d, e, f, g, h and i in table 3), the longest (composite f, g h and i)
corresponding to the variability of the signal at a 320 yr resolution and covering the whole 800 kyrs and the shortest (composite
a) corresponding to the variability of the signal at a 10 yr resolution and covering only the last 56 kyrs.



## 2.4 The EPICA ice core

To calculate the effect of isotopic diffusion with depth on the high resolution signal, we use the classical approach in which the initial isotopic signal is convolved with a Gaussian function G(z) of associated diffusion length $\sigma_z$ (Gkinis, 2011; Laepple, 2018; Gkinis et al., 2021):

$$G(z) = \frac{1}{\sigma_z\sqrt{2\pi}} \exp\left(\frac{-z^2}{2\sigma_z^2}\right) \tag{1}$$

Where z is the depth along the ice core and $\sigma_z$ is the diffusion length.

We quantify the amplitude decay of the signal between the initial amplitude $A_0$ and the measured amplitude at a certain depth as described in Johnsen et al., (2000) and Gkinis et al., (2021) for given period λ with the following equation:

$$\frac{A}{A_0} = \exp\left(-2\left(\frac{\pi\times\sigma_z}{\lambda}\right)^2\right) \tag{2}$$

For our purpose, the diffusion length along the EDC ice core is calculated by considering the firn diffusion (i.e. due to water vapor diffusion in the open porosity) and the ice diffusion (i.e. due to water molecular diffusion in the ice matrix).

We used two different estimates for the firn diffusion length, $\sigma_{firn}$, along the EDC ice core. In a first approach, we assumed a constant $\sigma_{firn}$ all along the EDC ice core and take the value of 0.07 m estimated by Johnsen et al. (2000) for EDC. In a second refined approach, we considered a changing $\sigma_{firn}$ between interglacial and glacial periods as described in (Gkinis et al., 2021).

This leads to a $\sigma_{firn}$ varying between 0.075 m in interglacial period to 0.065 m in glacial period. The ice thinning, S, also affects the visible effect of firn diffusion length along the ice core so that the thinned firn diffusion length should be $\sigma_{thinned\_firn} = S\times\sigma_{firn}$. In this study, for consistency, we used the thinning function for the EDC ice core corresponding to the AICC2012 chronology (Bazin et al., 2013).

The ice diffusion depends on the thinning and the temperature. The following formulation permits to calculate the diffusion length associated with ice diffusion, $\sigma_{ice}$, as a function of age (and also depth) of the ice (Gkinis et al., 2011) :

$$\sigma_{ice}^2(\tau) = S(\tau)^2 \int_0^\tau 2D(t)S(t)^{-2}\,dt \tag{3}$$

With S the thinning of the ice layers at the considered age τ. In order to estimate the ice diffusion coefficient D(t), we use the classical formulation of Ramseier (1967):

$$D = D_0 \times \exp\left(\frac{-Q}{RT}\right) \tag{4}$$

with $D_0 = 9.13$ cm$^2$/s and Q = 59.820 kJ/mol. At -50°C, D is equal $8.866.10^{-14}$ cm$^2$/s and at -10°C, D is equal to $1.1993.10^{-11}$ cm$^2$/s, T represents the ice temperature from the borehole.

The total calculated diffusion length expected to be measured in the ice core could then be estimated using the diffusion length associated with the firn diffusion and the diffusion length associated with ice diffusion in a quadratic addition, so that:



$$\sigma_{z=}\sqrt{(\sigma_{ice}^2 + \sigma_{thinned\_firn}^2)} \tag{5}$$

The increase of the diffusion length for increasing depth in the ice core is shown in Figure S1. It is mainly due to the increase in temperature. The borehole temperature indeed evolves almost linearly from -53.5°C to -2.6°C along the 3255 m ice core (Buizert et al., 2021). The variation of the calculated diffusion length around 3000 m is explained by the variability of the thinning function (Dreyfus et al., 2007).

## 3. The EPICA ice core


Because $\delta^{18}O$ and $\delta D$ measurements were performed over a long period in different institutes using different methods, we checked the coherency of the different datasets in two different ways: 1/ comparison of the same samples measured by different techniques on different periods and 2/ comparison of the low resolution measurements (55 cm resolution) with a 5 points average of the high resolution measurements (11 cm resolution).

First, we used the new CRDS technique in 2019-2020 to measure two sets of samples already analysed within the period 2004-2010 by uranium reduction for $\delta D$ on MIS 5.5 (1670-1693 m) and by $H_2O$-$CO_2$ equilibration for $\delta^{18}O$ (1670-1793 m). Figures 2 and 3 provide two examples of analyses performed on all overlapping intervals. Additional comparisons of new vs old data are also presented in the supplementary material sections (Figure S5 and S6). The difference between the old and the new $\delta D$ series (Figure 2) seems to depend on the absolute value for $\delta D$ (negative difference for low $\delta D$ values). This is confirmed

through a statistical test on the correlation between the absolute value of $\delta D$ and the $\delta D$ difference between the two series of measurements leading to a Pearson coefficient of 0.13 and a p-value of 0.003. Such an isotopic-dependent feature may arise from a possible calibration effect despite the fact that exchanges of home water standards and regular intercalibrations were performed between the laboratories measuring water isotopes of the EDC ice cores.

Despite such tendency in the $\delta D$ differences between the two series, the absolute value of the difference remains small. We

use a Welch t-test to show that the- old and new time series have equal means at a 99.9 % confidence level (t=3.5, N=1000) with respect to the experimental margin of errors.

Finally, the distribution of the differences between the first and the new $\delta D$ measurements is not Gaussian and not centred around 0. Still, this distribution is narrower ($2\sigma = 0.8$ ‰ when fitted by a Gaussian curve) than a Gaussian distribution with $2\sigma = 1.4$ ‰ associated with the classical analytical uncertainty of the $\delta D$ measurements. The analytical uncertainty associated

with the CRDS measurements series has been evaluated from the analysis of the difference between the same samples (1000 samples, which represent 10 % of the whole series) measured twice, 1 to 3 months apart. The distribution of the difference between duplicated analyses of the same samples with the same method is Gaussian with 33 % of the $\delta D$ difference being higher than 0.7 ‰. We conclude that both $\delta D$ series are comparable.

In parallel, no dependence on $\delta^{18}O$ values is observed for the distribution of the differences between the old and new $\delta^{18}O$

series. The standard deviation of the series of difference between old and new $\delta^{18}O$ measurements ($2\sigma = 0.2$ ‰) is smaller





than the classical analytical uncertainty of the $\delta^{18}O$ measurements by CRDS ($2\sigma = 0.4$ ‰)  (Figure 3). A statistical test was made on the correlation between the absolute value of $\delta^{18}O$ and the $\delta^{18}O$ difference between the two series of measurements leading to a Pearson coefficient of 0.0049 and a p-value of 0.9. In addition, we did a Welch t-test of the equality between two averages in $\delta^{18}O$ to know if the difference between the old and new $\delta^{18}O$ data is significantly different within the experimental

margin of error. When doing so, the result shows that the two series have equal means at a 70 % confidence level (t=0.557, N=1000).

Second, we compared low (55 cm) and high resolution (11 cm) $\delta^{18}O$ series after gathering the five 11 cm neighbour samples (Figures S2 to S4). The difference between the two timeseries is $0.008 \pm 0.001$ ‰ (Figure S4). The comparison between low and high resolution series for $\delta D$ was already performed by (Pol et al., 2011, 2014). In the 2011 paper, the coherency between

55 cm and 11 cm samples was studied through the calculation of the average signal over five 11 cm data. They observed that the signal from the 55 cm samples is similar to the average signal with however a lower statistical accuracy ($1\sigma = 0.5$ ‰) than the average signal ($1\sigma = 0.$ 23 ‰).

The two comparisons performed above lead to the conclusion that the different $\delta^{18}O$ and $\delta D$ EDC datasets gathered here and displayed on Figure 1 are coherent. It is thus reasonable to merge all the datasets together and create a unique high resolution

time serie containing all data obtained within different laboratories at different periods and with different techniques.

## 4. Results and discussion

### 4.1 Recorded multi-decadal to multi-millenial isotopic variability over the last 800 kyrs

The compiled high resolution water isotope datasets on the EDC ice core is presented in Figure 1. For $\delta D$, 5 interglacial periods

have been analyzed at high resolution. For $\delta^{18}O$, we have a profile almost complete except MIS 7 and part of MIS 11. We use these times series to study the multi-decadal to millennial variability over the last 800 kyrs, extending the results of Pol et al., (2011, 2014), which focused on the evolution of the multi-decadal and multi-centennial variability during the Holocene, MIS 5 and MIS 11.

We applied the MRA decomposition on each of the 6 selected intervals (see Methods) and present decadal to multi-millennial

variability across the last 800 000 years (Fig. 4). We calculated the running standard deviation ($1\sigma$) on a 3 ka window and we use this value as an estimate of the level of variability. For the first MRA composite at 10yr resolution (a), we observe a stronger isotopic variability during the Holocene than during the Last Glacial Maximum (average $1\sigma$ of 0.46 ‰ and 0.24 ‰, respectively). The 20 yr variability (b) inferred from the second composite shows a globally uniform pattern over the last 150 kyrs. The 80 yr variability (d) is smaller during the interglacial periods ($1\sigma=0.18$ ‰) than during the glacial periods ($1\sigma=0.30$

‰) over the last 400 kyrs. The 160 to 640 yr variability (e to g) also shows a small decrease of variability over interglacial periods and decreasing variability for the oldest ice core sections. For the lower frequency variabilities (composite at 1280 and 2560 yr resolution, h to i) the amplitude of the variability envelope increases during glacial inception and glacial period with



a notable strong 2560 yr variability at the onset of MIS 9 ($1\sigma=1.13$ ‰ compared to an average of $1\sigma=0.20$ ‰ over the whole
series). The large centennial to multi-centennial water isotope variability in glacial periods can be linked to the succession of
the Antarctic Isotopic Maxima (AIM) during glacial periods (EPICA community members, 2004; Jouzel et al., 2007). Finally,
the decreasing amplitude of the signal variability toward old ages is probably the result of diffusion of water isotopes in firn
open porosity and ice crystal. While we can disentangle the effect of diffusion and climate driven isotopic variability for low
frequency signal and deep depth, the respective influences of diffusion and climate are less obvious to identify at shallower
depths and for high frequencies.

**4.2 Effect of isotopic diffusion on the recorded signal variability**

We evaluate the effects of diffusion on the isotopic signal recorded in ice core records by computing the decrease of Holocene
variability from equation (2). The calculated $A/A_0$ signal amplitude is hence scaled for each MRA composites to the mean
amplitude of the variability of the MRA composite signal between 2 and 8 ka for each resolution (Figure 6).

As explained in the section "methods", we used the ice diffusion coefficient from Ramseier (1967) with 2 different estimates
for $\sigma_{firn}$. The different estimates of $\sigma_{firn}$ do not have a significant effect on the calculated amplitude of the variability (Figure
5).

Diffusion has the expected effect to decrease the amplitude of the variability of the isotopic signal for older and deeper ice
core sections. On the 10 yr series (a), diffusion dampens by half the amplitude of the recorded variability of the last glacial
period compared to the Holocene. The calculated amplitude of the variability due to diffusion is actually much smaller than
the recorded one which suggests that either the 10 yr isotopic variability during the last glacial period is larger than the 10 yr
variability during the Holocene or that measurement noise is dominating the 10 yr variability.

For the bottom part of the ice core, i.e. sections older than 600 kyrs, the diffusion model overestimates the damping of
centennial and multi-centennial variability compared to what is retrieved from the ice core isotopic composition. This
discrepancy calls for future reassessment of the isotopic diffusivity in the bottom part of the EDC ice core.


**4.2 The climatic variability at different timescales over the last 800 kyrs**

Combining our high resolution water isotopic records with frequency analysis and the impact of diffusion, we can suggest
some patterns for the decadal to millennial climate variability over the last 800 kyrs.

First, at the decadal scale, our findings can be interpreted as a larger variability during the last glacial period compared to the
Holocene. The analysis of (Jones et al., 2017) using water isotopic record on the WAIS Divide record in West Antarctica
supports this higher variability at the decadal scale during the last glacial maximum. In this high accumulation site, diffusion
is not affecting much the variability with a 4-15 yr periodicity and the higher water isotopic variability observed during this
period is interpreted as an increase in the strength of the teleconnections between the tropical Pacific and West Antarctica.
This increase should be related to the increase of the Northern Hemisphere leading to a shift in the location of the tropical





convection. The same pattern is observed for the 20 yr periodicity (Figure 5, panel b), i.e. the calculated diffused variability is smaller than the measured one during the last glacial period while there is a good agreement between diffused and measured signal over MIS 5e. For the 40 and 320 yr periodicity (Figure 5, c to f), the variability of the last glacial period is also higher than the diffused Holocene variability. It is also the case for MIS 6 for the 80 yr periodicity (Figure 5, d) and MIS 8 and 10 for the 160 and 320 yr periodicity (Figure 5 e to f). For these periods and frequency ranges, the impact of diffusion on the variability

is limited, and the isotopic signal in the ice core preserved. The multi-centennial variability increase during glacial periods can be related to the presence of AIM.

    Our analysis hence shows that there is a clear enhanced isotopic variability during glacial periods at the multi-decadal to multi-centennial timescale in the EDC ice core which could be attributed to climate variability. This result is in agreement with the findings of Rehfeld et al. (2018) using a worldwide data synthesis showing increased interannual to millennial climatic

variability during the last glacial maximum with respect to the Holocene at all latitudes with an increase of the variance by a factor of 2 in the high latitudes of the Southern Hemisphere, a result in agreement with output of coupled model simulations (Rehfeld et al., 2018).

    Second, while the effect of diffusion is important when we want to compare variability from one interglacial periods to another, it does not affect much the evolution of the recorded variability during the course of an interglacial period.

In their previous study focused on the warm phase of MIS 5 (115.5 to 132 ka), Pol et al. (2014) used wavelet analysis of the 11cm resolution δD record and evidenced three different phases with different level of variability. The first phase from 111 to 119 ka has a low orbital forcing context but the variability increases during the entry in glaciation, with centennial dominant periodicities. The second phase from 119 to 123 ka is a stable warm phase, warmer than the Holocene. The δD variability of the second phase is notably lower than the other phases, 3.7 ‰ compared to a 4.5 ‰ average. Finally, during the third phase

there is again a higher variability with dominant multi centennial periodicities between 123 and 133 ka.

    When doing a similar analysis with our MRA decomposition, we find similar variability of the high resolution signal (Figure 6 a), i.e. the maximum amplitude of the multi-decadal to multi-centennial variability of the signal is encountered over the optimum of MIS 5 (phase 3 in Pol et al.,( 2014), between 125 and 131 ka) and toward end of this warm period (phase 1 of Pol et al., (2014)). The minimum amplitude of the multi-decadal to multi-centennial variability of the signal is encountered between

119 and 123 ka (phase 2 in Pol et al., (2014)) when the δ¹⁸O and δD signals are on a plateau.

    Thus, during MIS 5, multi-decadal to multi-centennial variability of the water isotopic signal can be interpreted as climate variability at these multi-decadal to multi-centennial timescales.  It can be compared to the variability over the interglacial period of MIS 9 (~ between 325 and 338.5 ka) also characterized by a temperature optimum at its start. The amplitudes of the variability for the different MRA decompositions for the interglacial period of MIS 9 cannot be directly compared to ones over

MIS 5 because of the effect of diffusion and thinning (see figure 5). However, in figure 6 b we observe the same pattern than for MIS5:  higher amplitudes for the multi-decadal to multi-centennial variability are observed over the δ¹⁸O optimum (333 – 338 ka) and at the end of this warm period (321 – 326 ka) while the minimum amplitudes for the multi-decadal to multi-centennial variability is observed over the plateau of the interglacial period (326 – 332 ka). This result strengthens the





conclusion of Pol et al. (2014) that the climate over temperature optimum of interglacial periods may also be more variable at

the multi-decadal to multi-centennial timescale. A parallel can be drawn with the higher high frequency water isotopic variability observed during temperature optimum of the AIM of the last glacial period on the EDC ice core (Landais et al., 2015) since this temperature optimum at the beginning of the interglacial could also be the result of millennial scale variability (Past Interglacials Working Group of PAGES, 2016).

## 5 Conclusion

We presented a synthesis of new and published 11 cm resolution profiles of δD and δ$^{18}$O over the last 800 kyrs on the EDC ice core. We showed that the various water isotopic data measured by different laboratories and techniques over the last 20 years have coherent calibrations and homogeneous quality within analytical uncertainty. As a result, they can be combined and we provide here a homogenous and complete data series of high resolution water isotopes of the EDC ice core.

A MRA decomposition of the water isotopic record at temporal resolution varying between 10 and 2560 years shows that the

variability during glacial periods at multi decadal to multi centennial timescale is higher than variability of the Holocene and that the variability is enhanced over early temperature optimum during MIS 5 and 9. These results are not influenced by diffusion in the firn open porosity and in the ice matrix but the interpretation of high resolution δD and δ$^{18}$O profiles should take this effect into account. Finally, our study calls for further analyses for quantifying the diffusivity in EDC which is essential in the perspective of the BE-OI ice core.






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

**Acknowledgement**

This work is a contribution to the European Project for Ice Coring in Antarctica (EPICA), a joint European Science Foundation and European Commission scientific program, funded by the European Union and by national contributions from Belgium, Denmark, France, Germany, Italy, Netherlands, Norway, Sweden, Switzerland, and the United Kingdom. The main logistic support was provided by Institut Polaire Français Paul-Emile Victor and Programma Nazionale Ricerche in Antartide.

A.G. was supported by the European Research Council under the European Union Horizon 2020 Programme ERC ICORDA (817493).



**Figures**

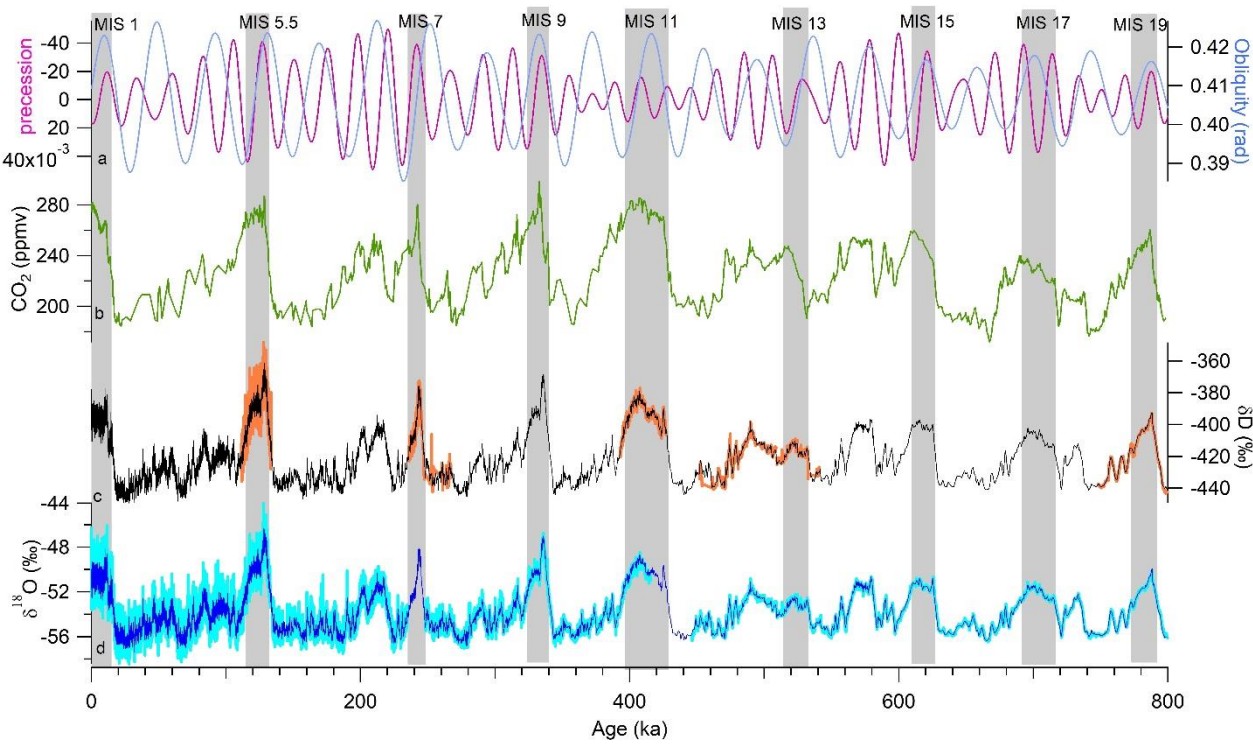

**Figure 1: High resolution water isotopic records over the last 800 kyrs on the EDC ice core. (a) precession (pink) and obliquity (blue) from Laskar et al., (2004); (b) Composite EDC and Vostok CO₂ record over the last 800 kyrs (Lüthi et al., 2008; Bereiter et al., 2015); (c) 11 cm (orange) and 55 cm resolution (black) of the EDC δD record; (d) 11 cm (light blue) and 55 cm resolution (dark blue) of the EDC δ¹⁸O record. All ice core records are presented on the AICC2012 scale (Bazin et al., 2013; Veres et al., 2013). Grey rectangles indicate the position of interglacial periods.**






| Place of measurements | Date | Age (ka) | Depth (m) | Resolution (m) | Method | 2σ (‰) | Reference |
|---|---|---|---|---|---|---|---|
| LSCE | 2010 | 112-134 | 1489-1756 | 0.11 | Uranium reduction | 1 | (Pol et al., 2014) |
| | 2002 - 2007 | 0-27 0-800 | 0-577 0-3189 | 0.55 | | | (Jouzel et al., 2001) (Jouzel et al., 2007) |
| | 2021 | 235-245 | 2253-2308 | 0.11 | | 1 | This study |
| | 2011 | 392-427 | 2694-2779 | 0.11 | | 1 | (Pol et al., 2011) |
| | 2010 | 747-801 | 3146-3189 | 0.11 | | 1 | (Pol et al., 2010) |
| | 2019 | 127-128.5 129.2-133 133.8-138 | 1670-1693 1704-1748 1756-1782 | 0.11 | CRDS spectroscopy analyser | 1.4 | This study |
| | 2019 | 245-267 | 2309-2372 | 0.11 | | 1.4 | This study |
| | 2019 | 451-542 | 2799-2913 | 0.11 | | 1.4 | This study |
| | 2019 - 2020 | 450-802 | 2772-3035 3044-3190 | 0.55 | | 1.4 | (Landais et al., 2021) |

**Table 1: Summary of available δD measurements on the EDC ice core and associated analytical methods. 2σ values come from instrumental measurement uncertainty as provided in the original studies.**




| Place of measurements | Date | Age (ka) | Depth (m) | Resolution (m) | Method | 2σ (‰) | Reference |
|---|---|---|---|---|---|---|---|
| LSCE | 2019 | 127-128.5<br>129.2-133<br>133.8-138 | 1670-1693<br>1704-1748<br>1756-1782 | 0.11 | CRDS spectroscopy analyser | 0.4 | This study |
| | 2019 | 245-267 | 2309-2372 | 0.11 | | 0.4 | This study |
| | 2019 | 451-542 | 2799-2913 | 0.11 | | 0.4 | This study |
| | 2019-2020 | 450-802 | 2772-3035<br>3044-3190 | 0.55 | | 0.4 | (Landais et al., 2021) |
| University of Copenhagen | 2001-2010 | 0-3<br>3-3.6<br>7-9<br>9.3-34.2<br>34.5-60<br>60-115 | 6.6-120<br>120-134<br>234-288<br>290-656<br>659-946<br>946-1528 | 0.11 | CO$_2$ equilibration | 0.14 | (Gkinis 2011)<br>(Landais et al., 2015)<br>(Gkinis et al. 2021 b) |
| | 2021 | 116-120<br>121-124<br>125-128<br>129-133<br>134-142<br>248-415<br>543-802 | 1539-1583<br>1594-1638<br>1649-1693<br>1704-1748<br>1759-1803<br>2317-2756<br>2794-3190 | | | | |
| University of Trieste<br>University of Parma | 2001<br>2004<br>2010<br>2021 | 0-27<br>0-44.8<br>0-140<br>0-800 | 0-590<br>0-787<br>0-1790<br>0-3190 | 0.55 | | 0.2 | (Stenni et al., 2001)<br>(Stenni et al., 2004)<br>(Stenni et al., 2010)<br>(Landais et al., 2021) |

**Table 2: Summary of available δ¹⁸O measurements on the EDC ice core and associated analytical methods. 2σ values come from instrumental measurement uncertainty as provided in the original studies.**





| Interval \\ Decom Positions | Interval 1 (0-56 kyrs) | Interval 2 (56 - 144 kyrs) | Interval 3 (144 - 305 kyrs) | Interval 4 (305 - 420 kyrs) | Interval 5 (420 - 651 kyrs) | Interval 6 (651 - 800 kyrs) |
|---|---|---|---|---|---|---|
| MRA 1 | 10 (a) | 20 (b) | 40 (c) | 80 (d) | 160 (e) | 320 (f) |
| MRA 2 | 20 (b) | 40 (c) | 80 (d) | 160 (e) | 320 (f) | 640 (g) |
| MRA 3 | 40 (c) | 80 (d) | 160 (e) | 320 (f) | 640 (g) | 1280 (h) |
| MRA 4 | 80 (d) | 160 (e) | 320 (f) | 640 (g) | 1280 (h) | 2560 (i) |
| MRA 5 | 160 (e) | 320 (f) | 640 (g) | 1280 (h) | 2560 (i) | |
| MRA 6 | 320 (f) | 640 (g) | 1280 (h) | 2560 (i) | | |
| MRA 7 | 640 (g) | 1280 (h) | 2560 (i) | | | |
| MRA 8 | 1280 (h) | 2560 (i) | | | | |
| MRA 9 | 2560 (i) | | | | | |

**Table 3: Time resolution of the different MRA decomposition for specific intervals (0-56, 56-144, 144-305, 305-420, 420-651, 651-800 kyrs). Letters a, b, c, d, e and f represent segments that have the same time resolution and can be combined.**





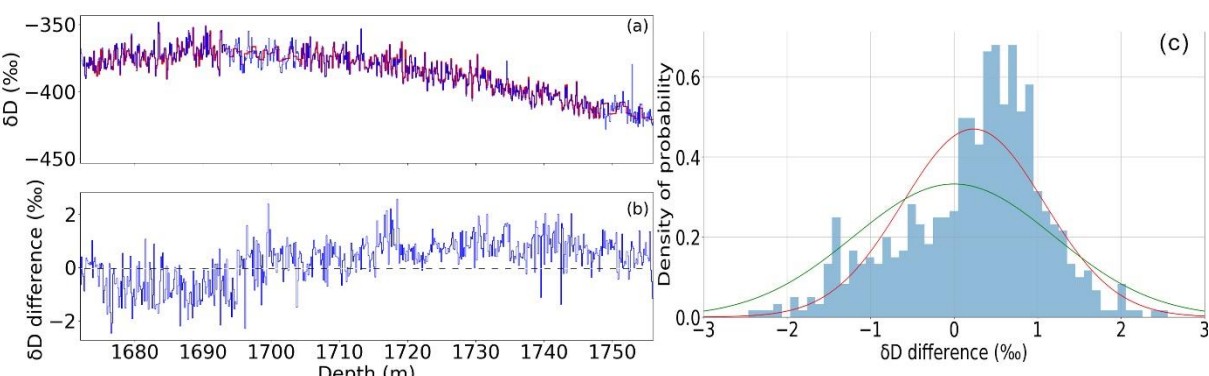

**Figure 2: (a) Evolution with depth of δD measurements over Termination 2 performed in 2010 at LSCE with the Uranium reduction method (Pol et al., 2014) (blue) and δD measurements performed in 2019 by CRDS at LSCE (red). (b) Difference between the δD values measured in 2010 and 2019. (c) Probability Density Function for the difference between the first (Uranium reduction) and the new (CRDS) δD measurements. A Gaussian curve (red) is fitted to the data. A gaussian curve (green) is displayed with the standard deviation equal to the classically displayed 1σ uncertainty of δD measurements with CRDS method at LSCE (1σ = 0.7 ‰).**

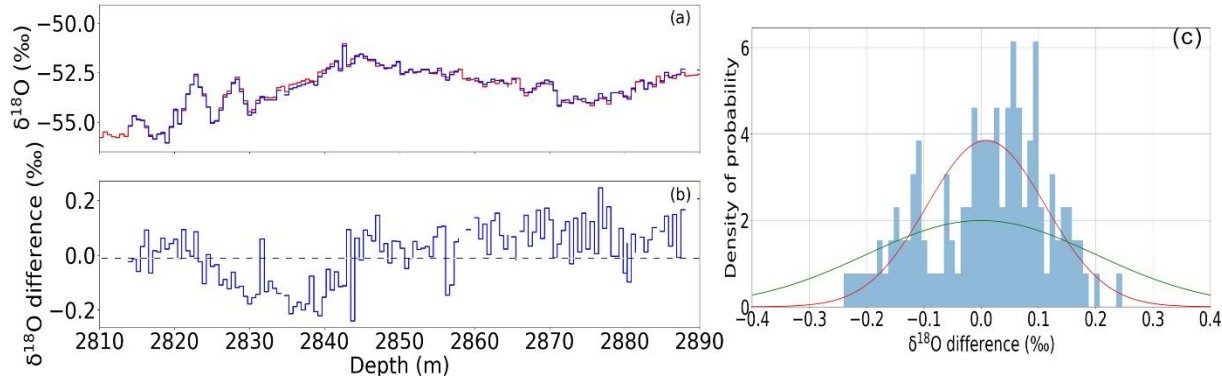

**Figure 3: (a) Evolution with depth of δ¹⁸O measurements over Termination 6 performed in 2010 at the University of Triestre with CO₂ equilibration method (blue) and δ¹⁸O measurements performed in 2019 by CRDS at LSCE (red). (b) Difference between the δ¹⁸O values measured in 2010 and 2019. (c) Probability Density Function for the difference between the old (University of Triestre) and the new (LSCE) δ¹⁸O measurements. A gaussian curve (red) is fitted to the data. A gaussian curve (green) is displayed with the standard deviation equal to the classically displayed 1σ uncertainty of δ¹⁸O measurements by CRDS at LSCE (1σ = 0.2 ‰).**



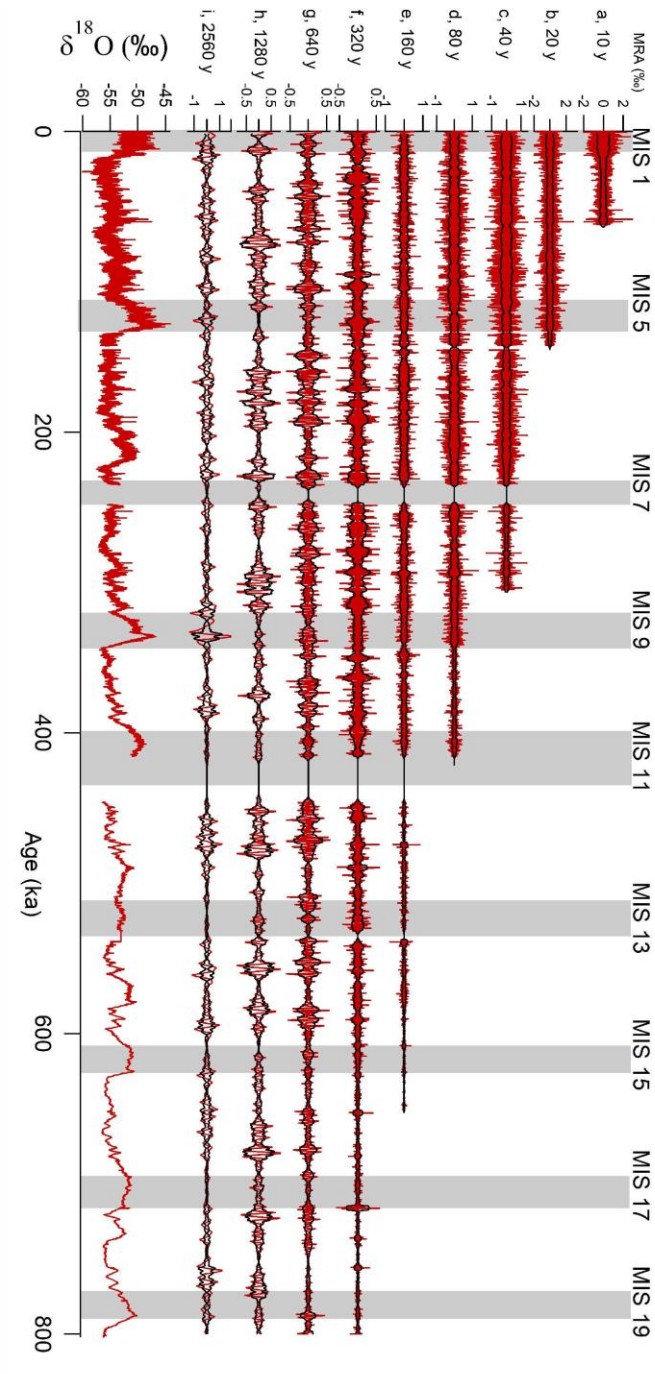

**Figure 4: Contribution to the original δ¹⁸O signal (red) of the MRA composites of resolution 10 (a), 20 (b), 40 (c), 80 (d), 160 (e), 320 (f), 640 (g), 1280 (h) and 2560 years (i). Marine Isotope Stage intervals are marked in grey bars. The black envelop presents the running standard deviation (1σ) on 3 kyr windows.**





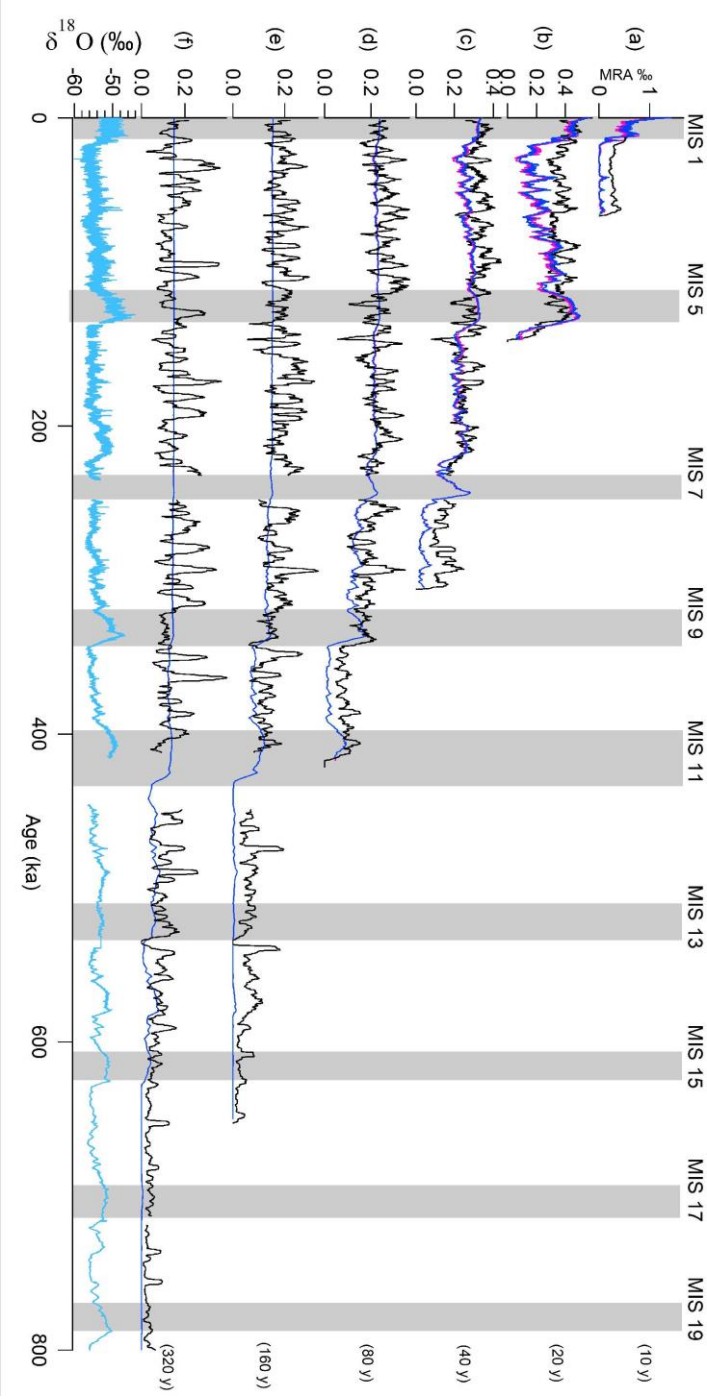

**Figure 5: High resolution record (light blue) and comparison of its variability (3 ky standard deviation, black) to the variability (3 kyr standard deviation) of the diffused Holocene signal for the different periods (10, 20, 40, 80, 160, 320 years for panels (a) to (f)). The diffused Holocene signal has been calculated using two σfirn estimates, one constant σfirn of 7 cm (dark blue), and one variable σfirn equal to 6.5 cm in glacial period and 7.5 cm in interglacial period (pink).**




490



**Figure 6: Contribution to the original δ¹⁸O signal (red) of the MRA composites of resolution 20 (b), 40 (c), 80 (d), 160 (e), 320 (f) and 640 (g) years (i) for MIS 5 (left - a) and MIS 9 (right - b). The black envelop presents the running standard deviation (1σ) on 3 kyr windows. The red rectangles indicate periods with enhanced variability and the blue rectangles indicate periods with reduced variability.**

495