# Peer review of "Sub-millennial climate variability from high resolution water isotopes in the EDC ice core"

_EGUsphere, 2022_

## Author Response (AR1)

Antoine Grisart
Laboratoire des Sciences et de l'Environnement
91190 Gif sur Yvette, France

Editor of Climate of the Past

Dear Pr Nancy Bertler,

Many thanks for your report. We corrected the manuscript according to the comments provided by the two referees. We have also precised in the answer to the reviewers the modification in the text for each occurrence and indeed, some indications were missing in the responses posted on the website before. We checked as well the consistency for the different notations
For the data availability we have send the file to Pangaea and hope to have the link soon to include it in the paper. Still, if it takes too much time, we will also post the new data on Zenodo to get a link in the final manuscript if accepted.

We hope that you will find this manuscript improved and suitable for publication in climate of the past,

On the behalf of all co-authors,
Antoine Grisart

Dear Andrew Moy,

Thank you very much for your review which will improve the manuscript. We corrected the document according to your suggestions and you'll find below all the modifications we propose.

Antoine Grisart et al.
* * *
Page 1, Line 14: Suggest changing '800 000 years' to '800,000 years'

>>> Done

Page 1, Line 15; The mentioning diffusion here, also requires the mentioning of water isotopes (d18O, dD).

>>> We replaced the sentence by "A unique opportunity to investigate decadal to millennial variability during past glacial and interglacial periods is provided by the high resolution water isotopic record ($\delta^{18}$O and $\delta$D) available for the EDC ice core"

Consider changing this sentence to 'The high resolution (11 cm) water isotopic record (d18O and dD) is available for the EDC ice core and accounting for water isotopic diffusion provides a unique opportunity to investigate decadal to millennial variability during past glacial and interglacial periods'.

>>> Replaced the original sentence by the following:

"A unique opportunity to investigate decadal to millennial variability during past glacial and interglacial periods is provided by the high resolution (11 cm) water isotopic record ($\delta^{18}$O and $\delta$D) available for the EDC ice core, accounting for water isotopic diffusion"

Page 1, Line 15: Change 'provide' to 'provides'

>>> Done

Page 1, Line 16: The use of the wording 'high resolution' can be sometimes be ambiguous to some depending on the site (e.g. inland or coastal site) and also depending on the accumulation rate. Also, the 11cm is reference to the sample resolution, and something like sample resolution for CFA - at millimetre resolution is also considered 'high resolution'? Suggest changing 'We present here a compilation of high resolution (11 cm) water isotopic records...' TO 'We present a continuous compilation of the EDC water isotopic record at a sample resolution of 11 cm that composed of 27,000 d18O and 7,920 dD measurements......'

>>> Beginning of the sentence has been replaced by the following:

"We present a continuous compilation of the EDC water isotopic record at a sample resolution of 11 cm which consists of …"

Page 1, Line 19; Consider changing 'We show that overlapping ..... homogeneous data set.' TO something like 'Here, we demonstrate that repeat water isotope measurements on the EDC ice core using different analytical methods on the same samples from different depth intervals are comparable within analytical uncertainty. From this comparison we combine EDC water isotope measurements to generate a high resolution (11 cm) data set over the past 800 kyrs.'

>>> We replaced the sentence by:

"Here, we demonstrate that repeat water isotope measurements on the same EDC samples from different depth intervals obtained using different analytical methods are comparable within analytical uncertainty. We thus combine all available EDC water isotope measurements to generate a high resolution (11 cm) data set over the past 800 kyrs"

Page 1, Line 27-29: The sentence 'Along air mass transportation, distillation......' needs to be explained better as you are trying to explain the use of 'water stable isotopes' in polar regions in a single sentence. E.g. - the loss of heavy isotopes - is some ways it would be good to mention what is a heavy or light isotope OR the oxygen and hydrogen isotope ratios?

>>> We added a sentence explaining briefly the isotope ratio: "$\delta^{18}O$ and $\delta D$ of water from ice core samples is classically measured with delta notation ($\delta$) expressing the variations of isotopic ratio of heavy to light isotopes in the water molecule (i.e. $^{18}O/^{16}O$ and D/H for $\delta^{18}O$ and $\delta D$). Along air mass transportation, distillation of moisture from the low latitude regions of evaporation to the polar regions leads to a preferential loss of heavy isotopes ($H_2^{18}O$ and $HD^{16}O$ vs $H_2^{16}O$)."

Page 1, line 27; 'Water isotopes' are not actually a 'tool to reconstruct past temperatures in polar regions'. Water stable isotopes (d18O, dD) are proxy records that can be used to reconstruct past temperatures'. Consider changing 'Water isotopes in ice cores (d18O, dD) are valuable tools to reconstruct past temperatures in polar regions' TO 'Water stable isotopes (oxygen, d18O; and hydrogen, dD) in ice cores are valuable proxy records that can be used to reconstruct past temperatures in polar regions'.

>>> Done following the suggestion

Page 1, Line 27; Consider changing 'Water isotopes in ice cores (d18O, dD)' TO 'Water isotopes in ice cores (oxygen, d18O; hydrogen, dD)'

>>> Done

Page 2, Line 36; Please be consistent with using 'kyrs' and 'ka'. For example - 'kyrs' is used here and at Page 2, Line 43-44, 'ka' is used.

>>> We differentiate ka to kyrs as they have different significance as suggested already by the editor before the discussion phase. We used 'ka' when expressing a duration and 'kyrs' for a date.

Page 2, Line 37; Suggest changing 'displayed .....' to 'measured at ~4 m resolution detailing dD variations over 8 glacial - interglacial cycles (EPICA Community members, 2004)'.'

>>> Done but replacing the 'detailing' with 'unveiling' in the sentence suggested

Page 2, Line 39; It might be a good idea to provide more info on 55cm? Are the bag samples composed of 55cm pieces of EDC or are the samples taken at 55cm intervals?

>>> It has been explained with: "continuous 55 cm pieces of the EDC ice core"

Page 2, Line 39; Delete 'systematic'.

>>> Done

Page 2; Line 40; The sentence 'In the following years, some studies ... climate variability.' as this is repeated in the next sentence.

>>> Sentence deleted

Page 2, Line 47; 'affecting the signal'? What is meant by the signal?

>>> This sentence was already not clear. It has been rewritten as follows: "Pol et al. (2014) used the high resolution water isotopic signals over MIS5 and 11 interglacial periods to estimate the relative variations of decadal to centennial climate variability during these interglacial periods with respect to the Holocene's (Pol et al., 2011; 2014)."

Page 2, Line 60; delete 'while we know the'

>>> Done

Page 2, Line 61; Consider changing 'we lack documentation' TO 'there is limited evidence in high resolution climate variability.....'

>>> We added 'available to document' after the 'evidence' in the sentence.

Page 3, Line 69; Change '3147 - 3190 m' TO '3,147 - 3,190 m'

>>> Done

Page 3, Line 70; Consider changing 'because' TO 'due'

>>> Replaced including the right preposition 'because of' by 'due to'

Page 3, Line 84; Change '3 233 m' to '3,233 m'

>>> Done

Page 3, Line 85; Consider changing 'around' to 'ca.'

>>> Done

Page 3, Line 85-86; change 'water equivalent.yr-1' to 'water equivalent yr-1'

>>> Done

Page 3, Line 87; Suggest re-wording 'on the Dome C where the ice was supposed to be the less deformed' and providing a reference?

>>> Changed by 'where the ice flow was small' and added the reference of (EPICA community members, 2004)

Page 3, Line 88; Suggest re-writing 'The drilling project was conducted ......' to 'The EDC drilling project started in 1996 and was completed in 2004. In 1999, a second ice core (EDC2) was drilled from the surface due to the drill for EDC1 being stuck at depth of 788 m. Bedrock was reached in 2004 at a depth of 3,190 m. From here onwards, we refer to EDC1 and EDC2 as the EDC ice core'

>>> Done following the suggestion

'After drilling and core logging, the EDC ice core was cut into 55 cm long sections and each section was further cut longitudinally on site for several measurements (e.g. water isotopes, physical properties, 10Be, chemistry, and gas analysis). The archival piece (~ one quarter of the section) was stored in polystyrene boxes in the EPICA snow-cave at the Concordia station at -50°C).'

>>> Replaced from the original sentence: 'After drilling and logging, the ice core was cut in 55 cm long parts. 55 cm sections were then cut longitudinally on site for several measurements (water isotopes, physical properties, $^{10}$Be, chemistry, gas). An archive piece (~ one quarter of the section) is stored in polystyrene boxes in the EPICA snow-cave at the Concordia station at -50°C.'

Page 3, line 97; I am assuming 'EDC' here means the 'EDC2' ice core? Although, please see the earlier comment 'from here onwards, we refer to EDC1 and EDC2 as the EDC ice core' as this should cover this off now?

>>> We added the sentence "From here onwards, we refer to EDC1 and EDC2 as the EDC ice core."

Page 3, Line 97; I suggest changing 'continuous' to 'contiguous'? Using 'continuous' might be taken as 'continuous flow analysis (CFA)'. Even though the EDC analysis is on samples at 55cm - it isn't really 'continuous' in terms of the meaning around CFA?

>>> The sentence has been rewritten as: "Two types of contiguous samples were dedicated for the analyses of water isotopes on the EDC ice core."

Page 3, line 98; consider changing 'Another section (stick with 2*1cm cross section)... TO 'The second was a 55cm length stick with a 2 cm2 cross section that was cut into 11cm length samples. Each sample was placed in a sealed plastic bag and stored at -20°C prior to being melted and transferred into plastic bottles that were kept at -20°C.'

>>> Done

Also - are the plastic bags 'whirlpak' or similar that are tightly sealed?

>>> We are using a plastic sheath cut to obtain a plastic bag at the right dimension and then, the bag is thermally sealed. We modified the text by "Each sample was placed in a plastic sheath cut to obtain a plastic bag at the right dimension and then, the bag is thermally sealed. The sample is stored at -20°C during a few months prior to being melted and transferred into plastic bottles that were kept at -20°C."

Page 4, Line 102; Considering writing this section to read something like 'Several analytical techniques have been used to measure d18O and dD on the EDC1 and EDC2 ice cores (Tables 1 and 2). Initial analytical techniques included uranium reduction method for dD (Vaughn et al., 1988); CO2 - H20 equilibrium method for d18O (Myer et al., 2000); with the most recent method to determine d18O and dD on the EDC2 ice core using cavity ring down spectroscopy (CRDS) (Kerstel and Gianfrani, 2008; Busch and Busch, 1999). The analytical precision for each method are comparable where 2σ values range between 1 and 1.4 ‰ for dD and between 0.1 and 0.4 ‰ for d18O (Table 2).

>>> Done

Page 4, Line 9; Page 5, Line 130, and Page 6, Line 165 - Please clarify the 'subheadings' used at 2.3, 2.4, and 3. as they are all 'The EPICA ice core'.

>>> This has been corrected. Sorry for the problem during final editing.

Suggest changing:

'2.3 The EPICA ice core' to '2.3 Discrete wavelet analysis' OR 'Multi resolution analysis (MRA)'?

>>> We chose the "Multi resolution analysis (MRA)" subtitle.

'2.4 The EPICA ice core' to '2.4 Isotopic diffusion'

>>> We used "Effect of isotopic diffusion" as subtitle.

'3. The EPICA ice core' to '3. Coherency of different analytical measurements'?

>>>Done

Page 4, Line 111; 'Delete 'With thus aim' and consider 'We produced a multi resolution analysis (MRA)....wavelet filter.'

>>> Done as suggested.

Page 4, Line 117; Consider changing 'The wavelet analysis needs to be applied on time intervals with a uniform resolution. Because we aim to keep....' TO 'The wavelet analysis needs to be applied on time intervals with a uniform sample resolution, and here we divide the EDC isotopic record on the AICC2012 age scale (add reference here) into six intervals. These include the youngest interval between 0 and 56 ka; where 11 cm corresponds to a 10 yr resolution; to the bottom of the core where the oldest interval between 651 and 800 ka; where 11 cm corresponds to a 320 yr resolution on the AICC2012 age scale (Table 3).'

>>> We modified the sentence by the following:

"As the wavelet analysis needs to be applied on time intervals with a uniform sample resolution, we divide the EDC isotopic record on the AICC2012 age scale (add reference here) into six intervals. These include the youngest interval between 0 and 56 ka (where the longest time span covered by 11 cm is 10 yr) to the bottom of the core with the oldest interval between 651 and 800 ka (where the longest time span covered by 11 cm is 320 yr) on the AICC2012 age scale (Bazin et al., 2013) (Table 3). Over each interval, we performed an interpolation with a uniform resolution corresponding to the longest time span covered by 11 cm of ice (i.e. interpolation at 10 yr between 0 and 56 ka, 20 yr between 56 and 144 ka, see details for all periods on Table 3)."

Again - please use 'kyr' or 'ka'

>>> See comment above. We are still using kyr and ka since they have different meanings.

Please reference the AICC2012 age scale.

>>> Done with the addition of the reference to (Bazin et al., 2013)

Page 5, Line 130; Consider changing '2.4 The EPICA ice core' TO 'Isotopic Diffusion'

>>> Done using: 'Effect of isotopic diffusion'

Page 5, Line 131; Consider changing 'To calculate the effect of isotopic diffusion....' TO 'The effect of isotopic diffusion with depth is convolved using a function G(z) of associated diffusion length σz (Gkinis, 2011; Laepple, 2018; Gkinis et al., 2021):'

>>> Done

Page 5, Line 157; Consider changing 'could then be' TO 'is'

>>> Done

Page 6, Line 161; Again, please better define the numbers with 'comma'. Suggest changing '3255 m' TO '3,255 m'.

>>>Done

Page 6, Line 162; Please change '3000 m' to '3,000 m'

>>>Done

Page 6, Line 165; Consider changing '3. The EPICA ice core' to '3. Coherency of different analytical measurements'?

>>>Done

Page 6, Line166; Consider changing 'Because d18O and dD measurements.....' TO 'Different analytical instruments and techniques have been used to determine d18O and dD in the EDC1 and EDC2 ice cores at different laboratories (Table 1). To determine the coherency of the different datasets, two different comparisons are made; (1) comparison of the isotopic values from the same samples measured by different analytical techniques; and (2) comparison of the 55 cm sample resolution data with the 11 cm sample resolution data using a 5-point average'.

>>> Changed as suggested

Also - is the 5 point average a 'moving average'?

>>> No, we averaged the 11 cm resolution on a 5-points window to compare it with the 55 cm resolution measurements on exactly the same window. We added this sentence to the new text: "We averaged the 11 cm resolution on a 5-points window to compare it with the 55 cm resolution measurements on exactly the same window."

Page 6, Line 170; Consider changing 'First, we used the new CRDS technique.....' TO start off with a new subheading '3.1 Comparison of isotopic data using different analytical techniques:  The CRDS analysis in 2019-2020 measured previously analysed samples from 2004-

2010; uranium reduction for δD on MIS 5.5 (1670-1693 m) and by H2O-CO2 equilibration for
δ*18O* (1,670-1,793 m).'

>>> We modified as suggested

dD comparison:

And consider also having sub heading for the dD comparison and d18O comparison? This
consideration would make reading this section of the manuscript easier. Understanding any
difference and the explanation for this difference will be critical for the manuscript.

>>> We sub headed the two parts with "3.2 δD comparison" and "3.3 δ18O comparison" as
suggested.

Page 6, Line 172; Consider changing 'Additional comparisons of new vs old data....' TO
'Additional comparisons of isotopic data measured by different analytical techniques on the same
samples are also presented in the ....'

>>> Done

Page 6, Line 173; Consider not using 'The difference between the old and the new'. Considering
changing 'The difference between the old and the new' TO 'The difference between analytical
techniques (Figure 2) .....'

>>> Done

Page 6, Line 177; Consider changing 'home water standards' TO 'internal laboratory water
standards'?

>>> Done

Page 6, Line 178; Can the isotopic difference be due to storage issues? For example - once
samples were initially analysed, were they re-frozen immediately after analysis? And did they
stay refrozen to ensure minimal evaporation?

>>> We precise this in the text using the following text: "The samples stayed refrozen between
the different measurements and they have been refrozen immediately after analysis. Tests have
been performed by storing low $\delta^{18}O$ and δD internal standards for several years in the freezer. In
some cases, but not systematically and not significantly compared to the analytical precision, a
small increase of $\delta^{18}O$ and δD could be obtained. In the comparison of the old and new record,
we do not observe a systematic increase of $\delta^{18}O$ and δD for the samples analysed recently
compared to the analyses performed 15 years ago so that we can unfortunately not give a solid
explanation for the small differences between the series of measurements."

Also - have repeat measurements using uranium reduction method for δD in 2004-2010 been
repeated in 2019-2020 OR is this analytical capability not available or viable now?

>>> We are sorry that there was a confusion on Table 1 for the δD data performed in 2021. They were measured by CRDS and not uranium reduction. Uranium reduction method is no more feasible now and δD can only be measured by CRDS now. Indeed, as indicated by the new Table 1, we compare data obtained using the older technic using uranium reduction with the lowest uncertainty with data obtained recently using the CRDS method with a higher uncertainty.

Page 6, Line 180; Change 'N=1000' TO 'N = 1,000'

>>> Done

Page 6, Line 182; The use of the wording 'first, new and old' can get somewhat confusing. Maybe considering upfront when the use of 'first, new and old' is used to actually define them? Or maybe this could be done in the Figure captions for Tables 1 and 2.

>>> We suggest using the year of measurements (2010 vs 2019) to replace 'old' and 'new', as implemented in the new text.

Page 6, Line 185; Consider changing '1000' to '1,000'

>>> Done

Page 6, Line 188; Just wondering how you can 'conclude that both dD series are comparable' with the dD difference between these repeat measurements that at 1 to 3 months apart'? If the 2-sigma difference 1.4 permile? Which is substantially larger than 2-sigma of 0.8 permile for the difference between first (chromium reduction) and the new (CRDS) measurements of the same samples for dD?

>>> The comparison of δD measurements of the same samples performed by CRDS 1 to 3 months apart leads to a gaussian repartition with a 2σ of 1.4 permil. When we do the same comparison between the measurements of the same samples performed by uranium reduction and CRDS, we find that the difference between the δD results is embedded within a gaussian curve with a 2σ of 0.8 permil. We thus conclude that the δD difference between the uranium reduction vs CRDS datasets is smaller than the uncertainty associated with CRDS measurements and thus that we can combine the different dataset if we consider a 2σ uncertainty of 1.4 permil on the final δD data.

The new text is now : "Still, this distribution is narrow and is encompassed within a Gaussian distribution with $2\sigma = 1.4$ ‰ associated with the classical analytical uncertainty of the δD measurements. Note that the analytical uncertainty associated with these CRDS measurements series has been evaluated from the analysis of the difference between the same samples (1,000 samples, which represent 10 % of the whole series) measured twice, 1 to 3 months apart. We thus conclude that the δD difference between the uranium reduction vs CRDS datasets is smaller than the uncertainty associated with CRDS measurements and thus that we can combine the different dataset if we consider a 2σ uncertainty of 1.4 ‰ on the final δD data."

The 2-sigma difference of 1.4 permile for repeat CRDS measurements is similar to the Gaussian dist. of the difference between first (chromium reduction) and the new (CRDS) measurements of the same samples for dD?

>>> cf answer to comments above. Note that we never did any chromium reduction, it is only uranium reduction or CRDS.

Page 6, Line 188; Has anyone considered completing repeat sample measurements for dD of the first (chromium reduction) with chromium reduction method today? This may not be in the scope of the manuscript - but if it has been completed - please mention something; or if there is a totally valid reason why it has not be completed - e.g. Cr method and mass spec no longer available?

>>> As mentioned above, uranium reduction is no more available. However, when uranium reduction has been replaced by CRDS measurements at LSCE, extensive series of comparison have been performed showing that there was an excellent agreement between the two methods within the uncertainty ranges of the instruments. We added the sentence in the new text "When uranium reduction has been replaced by CRDS measurements at LSCE, extensive series of comparison have been performed showing that there was an excellent agreement between the two methods within the uncertainty ranges of the instruments"

Page 6, Line 189; What is actually meant by 'no dependence'? Do you mean there is 'no significant statistical difference between d18O measurements completed using the CO2-equilibrium and CRDS method?

>>> Thank you for the suggestion. We have modified the sentence accordingly.

Page 7, Line 196; 'Consider changing 'N=1000' TO 'N=1,000'

>>> Done

Page 7, Line 197; What is meant by 'gathering'? Do you mean that you have calculated the isotopic average of five 11cm samples that overlap with the same sample depth as the 55cm samples?

>>> Yes, it is what we did and we changed the sentence accordingly. "Second, we compared low (55 cm) and high resolution (11 cm) $\delta^{18}O$ series after calculating the $\delta^{18}O$ average of five 11 cm samples that overlap with the same sample depth as the 55 cm samples (Figures S2 to S4)."

Page 7, Line 203; Consider changing 'The two comparisons performed.....' TO 'The two comparisons performed above suggest there is no signification statistical difference in the d18O and dD in the datasets compiled here (Figure 1).'

>>> Done

Page 7, Line 204; Consider deleting the sentence 'It is thus reasonable to merge all datasets....'.

>>> Here is the deleted sentence: 'It is thus reasonable to merge all the datasets together and create a unique high resolution time serie containing all data obtained within different laboratories at different periods and with different techniques.'

Page 7, Line 209; Consider changing 'The compiled high resolution.....' TO ' The compiled high resolution EDC water isotope record in present in Figure 1.'

>>> Done

Page 7, Line 209; The following sentences could be captured in the Figure 1 caption and hence probably don't need to be repeated here 'For δD, 5 interglacial periods have been analyzed at high resolution. For δ18O, we have a profile almost complete except MIS 7 and part of MIS 11. We use these times series to study the multi-decadal to millennial variability over the last 800 kyrs, extending the results of Pol et al., (2011, 2014), which focused on the evolution of the multi-decadal and multi-centennial variability during the Holocene, MIS 5 and MIS 11.

>>> This sentence is now removed.

Page 7, Line 215; Change '800 000 years' TO '800,000 years'
>>> Done

Page 7, Line 221-222; Change '1280 and 2560 yr' TO '1,280 and 2,560 yr'

>>> Done

Page 8, Line 223; Change 2560 yr' TO 2,560 yr'

>>> Done

Page 8, Line 224; Consider changing 'can be' TO 'is'

>>> Done

Page 8, Line 226; Is it actual old ages? Or is it towards 'larger time intervals'?

>>> It is old ages. It means that the deeper/older, the more diffuse. The text is now "Finally, the decreasing amplitude of the signal variability toward older ages is probably the result of diffusion of water isotopes in firn open porosity and ice crystal."

Page 8, Line 228; Deep depth? Or do you mean with 'increasing depth'?

>>> We replaced "deep depth" by "greater depth"

Page 8, Line 236; Maybe need to add a figure or table reference at the end of the sentence 'Diffusion has the expected effect to decrease the amplitude of the variability of the isotopic signal for older and deeper ice core sections (Figure or table?).

>>> The reference has been added to illustrate this idea : "Diffusion has the expected effect to decrease the amplitude of the variability of the isotopic signal for older and deeper ice core sections (Figure 4)."

Page 8, Line 242; Considering changing "bottom part' TO 'deepest' or 'oldest' sections (e.g. xxx depth or older the 600 ka')'

>>> Done

Page 8; Line 246; The subtitle '4.2 The climatic variability at different timescales over the last 800 kyrs' is not correct. This section is looking at the 'climate variability at different time intervals over the last 800 kyrs'. E.g. decadal, etc.

>>> Changed

Page 8, Line 252; Consider changing 'is not affecting much variability' TO 'diffusion has minimal affect on the variability........ (Jones et al., 2017)'.

>>> Done

Page 8, Line 254; What is meant by 'increase'? Do you mean 'The increase water isotopic variability.....' (maybe consider citing a reference to support this claim?).'

>>> This part was indeed not very clear and it has been rewritten with the addition to reference to Jones et al., 2018

"In this high accumulation site, diffusion has minimal effect on the variability with a 4-15 yr periodicity and the higher water isotopic variability observed during this period is interpreted as an increase in the strength of the teleconnections between the tropical Pacific and West Antarctica (Jones et al., 2018). Jones et al. (2018) invoke the expansion of the Northern Hemisphere ice sheets during the LGM leading to a shift in the location of tropical convection to explain these characteristics."

Page 9, Line 255; It might be a good idea to consider clarifying what is actually meant by 'the calculated diffused variability...'. I am assuming you mean 'the calculated water isotopic diffused variability....'.
>>> Thank you for this suggestion which was added

Page 9, Line 263; Consider deleting 'hence'

>>> Done

Page 9, Line 263; Consider adding a reference to a figure at the end of this sentence?

>>> "Figure 4" has been added as reference at the end of this sentence

Page 9, Line 22; Delete 'much'
>>> Done

Page 9, Line 270; Consider re-writing this sentence to something like 'A previous studies focused on the warm phase of MIS 5 (115.5 to 132 ka), where the wavelet analysis of the 11cm resolution δD record showed there were three different isotopic phases with different levels of variability (Pol et al., 2014).

>>> Done

Page 10; Line 295; Consider changing 'We presented' TO 'Here, we compiled and presented a EDC ice core water isotopic record (d18O and dD) using new and previously published 11 cm data spanning the last 800 kyrs.....'

>>> Done

Page 10, Line 297; 'Coherent calibrations'? Not sure what this actually means?
>>> Done in the suggested comment below

Consider this 'Our compilation and comparison work showed that water isotopic data measured by different laboratories and techniques over the last 20 years on the same samples show no significant statistical difference and are within analytical uncertainty. As a result, the EDC water isotope data is combined to produce a contiguous high resolution data set at mostly 11 cm sample resolution'.
>>> We modified the suggested sentence by:

"This compilation and the comparison performed between different series of measurements showed that water isotopic data measured by different laboratories and techniques over the last 20 years on the same samples display no significant statistical difference and are within analytical uncertainty. As a result, all the available EDC water isotope data are combined to produce a continuous high resolution dataset at mostly 11 cm sample resolution."

Page 10, Line 299; Consider changing '2560 years' to '2,60 years'
>>> Done

Page 15, Line 440; Consider changing figure caption as Figure 1 contains more than just the water isotopic record from EDC, and other features (precession and obliquity). Consider changing to 'EDC ice core, other palaeoclimate records and variations in Milankovitch cycles over the past 800 kyrs'

>>> Done

For Tables 1 and 2 - consider adding a 'comma' for the depths (e.g. consider changing '1489-1756' TO '1,489 - 1,756' and so on for the other depths.

>>> Done

Figure 2 - what is meant by 'evolution with depth'? Do you mean 'EDC dD measurements versus depth (m) over Termination 2, where measured completed in 2010 at LSCE (Uranium reduction method; Pol et al., 2014) (blue) and δD measurements completed in 2019 at LSCE (CRDS method) (red).

>>>Thank you for this suggestion. We rewrote as:

"a) EDC ☐D measurements versus depth (m) over Termination 2: in blue measurements completed in 2010 at LSCE (Uranium reduction method; Pol et al., 2014) and in red measurements completed in 2019 at LSCE (CRDS method)"

Figure 3 - please see the suggested comments on Figure 2 as these are similar.

"(a) EDC dD measurements versus depth (m) over Termination 6: in blue measurements completed in 2010 at the University of Triestre with $CO_2$ equilibration method and in red measurements completed in 2019 at LSCE (CRDS method)."

>> This has been changed accordingly

References: The following reference is listed in the reference section but it could not be found in the manuscript: Fisher D. A., Reeh, N., Clausen, H.B.: Stratigraphic noise in time series derived from ice cores. Annals of glaciology 7, 1985.

>>> It was placed line 67 in the introduction along with Laepple 2018.

Dear Referee #2,

Thank you very much for your review which will improve the manuscript. We corrected the document according to your suggestions and you'll find below all the modifications we propose.

Antoine Grisart et al.

The authors applied a multi resolution analysis (MRA) to identify the contribution of the decadal to multi-millennial signal variabilities to the overall isotopic variability. The MRA method needs to be applied on time intervals with a uniform resolution. Therefore, the authors should provide more details how they handled the fixed 11cm points into a uniform resolution within each of the 6 intervals. Certainly the 11cm covers a different temporal coverage along depth, given thinning, as well as change in accumulation rate.

>>> We transformed the fixed 11 cm points resolution depth scale into the AICC2012 age scale which is not constant neither. But then we interpolated at a fixed time interval according to the maximum resolution allowed. For example, we took the first 900 m of the EPICA ice core which corresponds to 0-56 ka. The maximum time resolution at 56 ka is 10 years. So we uniformly interpolated the 0-56 ka to 10 years resolution. At 144 ka, the time resolution is 20 years so we interpolated the 56-144 ka to 20 years resolution. And so on.

We have rewritten the text to better explain this point which was probably not clear enough: "As the wavelet analysis needs to be applied on time intervals with a uniform sample resolution, we divide the EDC isotopic record on the AICC2012 age scale (add reference here) into six intervals. These include the youngest interval between 0 and 56 ka (where the longest time span covered by 11 cm is 10 yr) to the bottom of the core with the oldest interval between 651 and 800 ka (where the longest time span covered by 11 cm is 320 yr) on the AICC2012 age scale (Bazin et al., 2013) (Table 3). Over each interval, we performed an interpolation with a uniform resolution corresponding to the longest time span covered by 11 cm of ice (i.e. interpolation at 10 yr between 0 and 56 ka, 20 yr between 56 and 144 ka, see details for all periods on Table 3)."

I agree with the authors that the old and new δ18O and δD EDC datasets are coherent, just wonder if a more objective test can be applied to confirm their conherence?

>>> This is a good question and we have indeed searched what were the best statistical tests adapted to our purpose. Here, we used both the Pearson-test and the Welch test to compare our different set of data. Note also that we did several kinds of comparisons to address the coherence between the different sets of data as explained on section 3 which has been reorganized following the comments of reviewer 1. In particular, we now explain our strategy at the beginning:

"To determine the coherency of the different datasets, two different comparisons are performed; (1) comparison of the isotopic values from the same samples measured by different analytical

techniques; and (2) comparison of the 55 cm sample resolution data with the 11 cm sample resolution data using a 5-point average."

Line 304: BE-OI stands for Beyond EPICA-Oldest Ice?

>>> Yes, it is. We replaced the acronym with the full name in the text.